# OverFlap PCR: A reliable approach for generating plasmid DNA libraries containing random sequences without a template bias

**Artis Linars, Ivars Silamikelis, Dita Gudra, Ance Roga, Davids Fridmanis**⊙*

Latvian Biomedical Research and Study Centre, Riga, Latvia

* davids@biomed.lu.lv

**Data Availability Statement:** All data are within the paper and its Supporting Information files. Cutadapt v1.15 is a software tool that removes adapter sequences from sequencing reads. It is

## Abstract

Over the decades, practical biotechnology researchers have aimed to improve naturally occurring proteins and create novel ones. It is widely recognized that coupling protein sequence randomization with various effect screening methodologies is one of the most powerful techniques for quickly, efficiently, and purposefully acquiring these desired improvements. Over the years, considerable advancements have been made in this field. However, developing PCR-based or template-guided methodologies has been hampered by resultant template sequence biases. Here, we present a novel whole plasmid amplification-based approach, which we named OverFlap PCR, for randomizing virtually any region of plasmid DNA without introducing a template sequence bias.

## Introduction

Since the emergence of recombinant DNA technologies and their employment in practical biotechnology for producing desired biological compounds (around the mid-1970s) [1, 2], researchers in this field have viewed improving employed proteins and creating new ones as the holy grail [3, 4]. Over the years, several researchers focused their attention on developing and utilizing novel strategies and methods for fast, efficient, and purposeful DNA sequence alterations that encode their protein of interest and insertion into the appropriate expression vector. As one would expect, these efforts have resulted in numerous approaches continuously being modified and improved. Although the list of developed methods is extensive and might confuse any newcomer to the field, there are two questions every researcher must answer before selecting an appropriate approach:

1. "Is it necessary to alter a protein-coding sequence at a specific site or multiple sites?" and

2. "Is it necessary to replace a selected codon with a specific random sequence?".

The site and residue-specific alterations represent the simplest modification groups. Thus, basic PCR- [3], megaprimer- [5], or whole plasmid- [6] based site-directed mutagenesis coupled with restriction-ligation- [1], Gibson assembly- [7], recombination- [8], or deoxyuridine-USER enzyme (Uracil-Specific Excision Reagent)- [9] based cloning into a selected vector is

available in the GitHub repository (https://github.com/marcelm/cutadapt/). SeqAn 2.4.0 is an open source C++ library of efficient algorithms and data structures for analysing sequences with a focus on biological data. It is available in the GitHub repository (https://github.com/seqan/seqan/). OpenCFU is a free software that should facilitate the enumeration of colony forming units (CFU) and render them more reproducible. It is available in the SourceForge repository (http://opencfu.sourceforge.net/).

**Funding:** This work was supported by European Regional Development Fund (ERDF) [Project No.: 1.1.1.1/16/A/055] The funders had no role in study design, data collection and analysis, decision to publish, or preparation of the manuscript.

**Competing interests:** The authors have declared that no competing interests exist.

usually employed. These methods are reliable and proven to provide excellent results in numerous publications. However, introducing residue-specific alterations at multiple sites is more challenging. The simplest solutions would involve several repetitions of single-site directed mutagenesis cycles, but this approach is time- and labor- consuming. Therefore, methods such as progressive PCR-based multi-site-directed mutagenesis [10], mutant strand synthesis by primer extension and ligation [11], homologous recombination-based multi-site-directed mutagenesis [12], Multichange ISOthermal (MISO) Mutagenesis [13] QuikChange multiple site-directed mutagenesis [14], and many others, including various modifications of already mentioned, [15–20] were developed. Despite these differing approaches, all these methods are just as reliable as single-site mutagenesis methods because the success in both cases is determined by the necessity to acquire a single clone with desired alterations.

Conversely, the success of randomization approaches is determined by whether the method can deliver a library of sequences with the desired level of diversity and the greatest possible number of variants. Thus, selecting an appropriate method and evaluating its limitations is paramount. When considering multi-site randomization, four types of approaches that deliver significantly different sequence libraries can be distinguished:

1. Random mutations at random positions: usually introduced by employing Error-Prone PCR [21–26] or *Escherichia coli* mutator strains [27–30].

2. Random combinations of multiple sites and residue-specific alterations: introduced with the help of split-mix-PCR [31], RECODE [32], OSCARR [33], or several other techniques.

3. Random recombination of multiple pre-existing DNA sequences: acquired through DNA shuffling [34–36], StEP [37, 38], RACHITT [39, 40], or ITCHY [41–44].

4. Random residues at multiple specific positions: usually introduced in a similar manner as residue-specific alterations at multiple sites. However, a library of oligonucleotides that randomizes the selected position is employed instead of a specific alteration containing oligonucleotides [45].

According to the literature, the first three methods are either already well-developed and used for many years without significant alterations (error-prone replication/amplification) or are constantly being redesigned at their core to provide libraries of greater quality and simplify the procedure (other two). However, regarding randomizing residues at multiple specific positions, recent significant developments have not been made because those single-site methods that serve as the basis for these approaches are sufficiently effective and reliable for introducing small (up to six nucleotides (2 aa codon)) randomizations at a single site [46], which in essence is the target number for most studies. The necessity for introducing larger-scale randomizations is also recognized, as it would be beneficial for examining larger protein motifs. However, it is rarely employed because it usually results in a library with significant sequence distribution bias towards the employed template, which may considerably decrease the number of functional variants that display a sufficient difference from the source and quantitative dominance of only several variants. The only suggested means for reducing this effect involved carefully designing the oligonucleotides [47] and fine-tuning the oligonucleotide annealing temperature [48], which is also recommended if six or fewer nucleotides are randomized.

Consequentially, this bias problem is also acute in single-site randomization because it employs the same methods used for site- and residue-specific alterations [23, 47, 49, 50]. As one can see from the aforementioned information, most of these are PCR- or other types of template-based methods. Thus, here, too, if the target motif is larger than a few codons, then oligonucleotides with the greatest similarity to the template will bind/anneal with greater

efficiency than others, and the whole process will result in a library with decreased sequence diversity. However, unlike in residue-specific replacement, there are several additional frequently used methods specifically designed to minimize this bias. The most prominent is cassette mutagenesis [51–53], where restriction/ligation is used to insert chemically synthesized, randomized sequences containing fragments into the target site. However, this method has a few technical drawbacks. First, there is the need for conveniently located restriction site/-s, and if these are not available, they have to be introduced using site-directed mutagenesis without (or minimally) disrupting the encoded aa sequence. Second, in our experience, the efficiency of two-fragment ligation is significantly lower than that of circularization. Thus, its employment limits the acquired library's diversity. Neme *et al.* [53] reported that this strategy reliably identified ~1000 unique variants per experiment.

The loop-out/loop-in method also addresses this bias issue [54]. Although this technique is based on classical PCR mutagenesis, undesired selectivity is controlled by excising the target site during the first round of mutagenesis and inserting a randomized sequence during the second. This approach should eliminate the bias towards the target site, but it might introduce a bias towards the adjacent and hairpin structure-forming sequences. As this method is infrequently employed, the literature on its drawbacks is scarce.

Researchers working with randomized libraries should also consider their quantitative characterization for both quality assessment and experimental purposes (acquired data might serve as point zero in experiments assessing changes in the relative abundance of clones). For studies carried out more than a decade ago, Sanger sequencing was almost the only available option. Researchers had to handpick individual colonies to acquire fragments with the inserted sequence. Since such a strategy was laborious, costly, and had a low throughput, it usually resulted in low data amounts. This limitation is largely overcome with the widespread availability of massive parallel sequencing technologies (also known as next-generation sequencing (NGS)) [15, 53, 55, 56]. However, we also observed that in several reports, particularly those that claimed to have developed a "novel and highly effective" randomization strategy, the NGS data were not presented [49, 57], indicating that applying the methodology to selected tasks should be reviewed critically, and these claims should be considered with caution.

Thus, in light of the presented information, we believe that specific multi- and single-site mutagenesis methods are still insufficiently developed, and there is room for significant improvement to create random sequence libraries without template biases. The underlying purpose of this methodological study was to construct an expression plasmid library to produce 18 aa long random peptides in *Saccharomyces cerevisiae* yeast to be used for screening novel biologically active molecules. During the course of these activities and while struggling with the described bias, we developed a novel approach for reliably introducing random nucleotide sequences within virtually any site of the plasmid, which we named "OverFlap PCR" to distinguish it from "Overhang PCR" and "Overlap PCR". While preparing this article, we intensely discussed methodological improvements, which we present in subsequent sections.

## Materials and methods

### Creating the *S. cerevisiae* compatible secreted peptide expression plasmid p426GPD-αfactor-αMSH

The p426GPD expression plasmid, kindly provided by Dr. Simon Dowell from GSK (Stevenage, UK), was supplemented with a coding sequence (CDS) for the *S.cerevisiae* α-factor secretion signal to enable the produced peptides to be secreted. We added an α-Melanocyte stimulating hormone (α-MSH) CDS to the 3' end of the secretion signal's CDS to create a

source plasmid that could be used as a positive control during the expression experiments. Both elements were inserted simultaneously in the following manner.

The secretion signal CDS was amplified by PCR from 0.05 μg of pPIC9K vector plasmid (Thermo Fisher Scientific, USA) using 2.5 U of Pfu polymerase (Thermo Fisher Scientific, Lithuania), 10 pmol of aFactor-BamHI-Fw forward primer (contained the *Bam*HI restriction site), 10 pmol of aFactor-aMSH-EcoRI-Rs reverse primer (contained the *Eco*RI restriction site and an α-MSH CDS) (Table 1). All the primers used in this study were purchased from Metabion GmbH, Germany. Other reaction reagents included 2 μl of 10x reaction buffer with MgSO$_4$ (Thermo Fisher Scientific, Lithuania), 4 nmol of each deoxyribonucleoside triphosphate (dNTP) (Thermo Fisher Scientific, Lithuania), and water to a final volume of 20 μl. The reaction was performed in a Veriti PCR thermal cycler (Applied Biosystems, USA) under the following conditions: 95˚C for 5 min followed by 40 cycles at 95˚C (15 sec), 63˚C (30 sec), and 72˚C (60 sec), finalized at 72˚C (5 min). We verified the success and purification of the acquired product from the template and primer dimers via preparative agarose gel electrophoresis, appropriate band excision, purification employing a GeneJET Gel Extraction Kit (Thermo Fisher Scientific, Lithuania), and elution in 18 μl of ultrapure water. Then, whole volume of the purified PCR product (αFactor-αMSH) and 2 μg of the p426GPD vector plasmid were cleaved with *Bam*HI (Thermo Fisher Scientific, Lithuania) and *Eco*RI (Thermo Fisher Scientific, Lithuania) restriction enzymes in *Bam*HI reaction buffer and 20 μl of the total reaction volume according to manufacturer's instructions. Following the restriction, both fragments (fαFactor-αMSH and v426GPD) were purified using a GeneJET PCR Purification Kit (Thermo Fisher Scientific, Lithuania) according to the manufacturer's instructions and eluted in 20 μl of ultrapure water. Subsequently, 1 μl of v426GPD and 7 μl of fαFactor-αMSH were mixed with 1 μl of 10x T4 DNA ligase reaction buffer and 5 U of T4 DNA ligase (Thermo Fisher Scientific, Lithuania), incubated for 1 h at 22˚C, and the whole reaction volume was transformed into chemically competent *E.coli* Dh5α strain cells (acquired from Invitrogen, USA and prepared according to Green & Rogers instructions [58]). The cells were seeded onto an ampicillin-supplemented lysogeny broth (LB) media petri dish and incubated overnight at 37˚C. We verified fragment insertion success through agarose gel electrophoresis visualization of the colony PCR products ((1xTaq reaction buffer, 17.5 nmol of MgCl$_2$, 4 nmol of each dNTP, 10 pmol of each M13-Fw and M13-Rs primer (Table 1), 0.5 U of recombinant Taq DNA Polymerase (Thermo Fisher Scientific, Lithuania), and water to a final volume of 10 μl;

**Table 1. Employed oligonucleotides.** Restriction sites are underlined, coding sequences are in *Bold-Italic*, and deoxy uridine is in **Bold**.

| Oligo Name | Oligo Sequence |
|---|---|
| aFactor-BamHI-Fw | 5'-GTGGATCCAAACAATGAGATTTCCTTCAATT-3' |
| aFactor-aMSH-EcoRI-Rs | 5'-TCGAATTCTTA***AACTGGTTTACCCCATCTAAAATGTTCCATAGAATAAGA***GTATGCTTCAGCCTCTCT-3' |
| M13-Fw | 5'-GTAAAACGACGGGCAG-3' |
| M13- Rs | 5'-CAGGAAACAGCTATGAC-3' |
| Ran-Pept-Fw | 5'-AGCATC**U**GAATTCGATATCAAGCTAGCTTC-3' |
| Ran-Pept-Rs | 5'-AGATGC**U**GCTTANNNNNNNNNNNNNNNNNNNNNNNNNNNNNNNNNNNNNNNNNNNNNNNNNNNGTATGCTTCAGCCTCTCT-3' |
| a-Factor-Fw | 5'-TACTATTGCCAGCATTGCTGC-3' |
| Ran-Pept-Lin-Rs | 5'-AGATGC**U**GCTTANNNNNNNNNNNNNNNNNNNNNNNNNNNNNNNNNNNNNNNNNNNNNNNNNNNGTATGCTTCAGCCTCTCTTTTT CTCGAGAGATACCCCTTCTTCTT-3' |
| NGS-RanSeq-Fw | 5'-CCTCTCTATGGGCAGTCGGTGATTGTTTTGCCATTTTCCAACA-3' |
| NGS-RanSeq-Rs-77 | 5'-CCATCTCATCCCTGCGTGTCTCCGACTCAGCGAAGCGATTCGATCGAAGCTAGCTTGATATCGAAT-3' |
| NGS-RanSeq-Rs-69 | 5'-CCATCTCATCCCTGCGTGTCTCCGACTCAGTTCAATTGGCGATCGAAGCTAGCTTGATATCGAAT-3' |

thermal conditions: 95˚C for 5 min followed by 45 cycles at 95˚C (30 sec), 55˚C (30 sec), and 72˚C (1 min 30 sec), finalized 72˚C (7 min)). Colonies that produced 1300 bp long DNA fragments were considered insertion-positive. Two positive colonies were inoculated in 6 ml of 2YT media, incubated overnight at 37˚C in a shaker, and the plasmid DNA was extracted using a GeneJET Plasmid Miniprep Kit (Thermo Fisher Scientific, Lithuania). We used Sanger sequencing with M13-Fw and M13-Rs primers, a BigDye™ Terminator v3.1 Cycle Sequencing Kit (Thermo Fisher Scientific, USA) (according to manufacturer's instructions), and a 3130/ 3130xl Genetic Analyzer (Applied Biosystems, USA) as the final means of verifying if the sequence fragment inserted. We selected the plasmid for further activities based on the quality of the acquired DNA sequences (i.e., only the clones that yielded high-quality chromatograms). S1 Fig. presents the map of the acquired plasmid.

### Peptide CDS randomization

**Randomization employing a modified whole plasmid amplification (WPA) strategy.** Our initial approach to randomizing the peptide CDS was based on a modified WPA strategy [6, 50]. The modifications included employing deoxyuridine-containing primers and USER enzyme mix (New England Biolabs, UK) to create sticky ends that would facilitate the circularization of the plasmid [9]. Fig 1 presents the principal scheme of the procedure.

The WPA strategy reaction mixture contained 1x Phusion U Multiplex PCR Master Mix (Thermo Fisher Scientific, Lithuania), 0.06 μg of p426GPD-αfactor-αMSH plasmid, 50 pmol of Ran-Pept-Rs primer (in essence was identical to the previously employed aFactor-aMSH-EcoR-I-Rs primer, but 54 random nucleotides replaced the α-MSH CDS and the 5' end contained an artificial sequence of six GC50% nucleotides that ended with uridine (Table 1)), 50 pmol of Ran-Pept-Fw primer (starting from the 5' end, it contained six nucleotides that were complementary to the reverse primer's artificial GC50% sequence and 23 nucleotides that were complementary to the sequence that follows immediately after the α-MSH CDS of the template plasmid), and water to achieve a final volume of 50 μl. The thermal conditions were as follows: 98˚C for 3 min followed by ten cycles at 98˚C (10 sec), 59˚C (30 sec), and 72˚C (3 min 30 sec), finalized at 72˚C (7 min). The success of the reaction was verified by agarose gel electrophoresis, and the acquired products were simultaneously treated with 2 U of USER enzyme mix and 20 U of *Dpn*I restriction enzyme (Thermo Fisher Scientific, Lithuania) for 1 h at 37˚C. The first reagent introduced nicks at the uridine site, releasing the first six 5' nucleotides and creating sticky ends. *Dpn*I exclusively cleaved methylated recognition sites, which are typically found only in bacteria-extracted DNA. Thus, in essence, only template DNA was sheared. DNA fragments were then purified using a GeneJET PCR Purification Kit and eluted in 20 μl of ultrapure water. Purified DNA (8 μl) was used for circularization employing T4 DNA ligase as described earlier. Subsequently, the whole reaction volume was transformed into chemically competent *E. coli* Dh5α strain cells (prepared inhouse employing the "Rubidium chloride competent cell protocol", which is available at https://mcmanuslab.ucsf.edu/protocol/rubidium-chloride-competent-cell-protocol and with an estimated transformation efficiency of $10^9$ CFU per 1 μg of pUC19 plasmid), which were seeded onto an ampicillin-supplemented LB media petri dish and incubated overnight at 37˚C. The following day, the number of transformation-positive colonies (and possible randomization clones) was estimated by petri dish imaging in a UVP Biospectrum® AC Imaging System (UVP, USA) and a colony count estimation by OpenCFU 3.9.0 software [59] (available at http://opencfu.sourceforge.net/). Then, all the colonies were washed off with 1 ml, inoculated in 5 ml of ampicillin-supplemented 2YT media, and incubated overnight at 37˚C in a shaker. Plasmid DNA extraction and sequencing were performed as described previously. However, a-Factor-Fw (Table 1) was used instead of the M13-Rs primer.

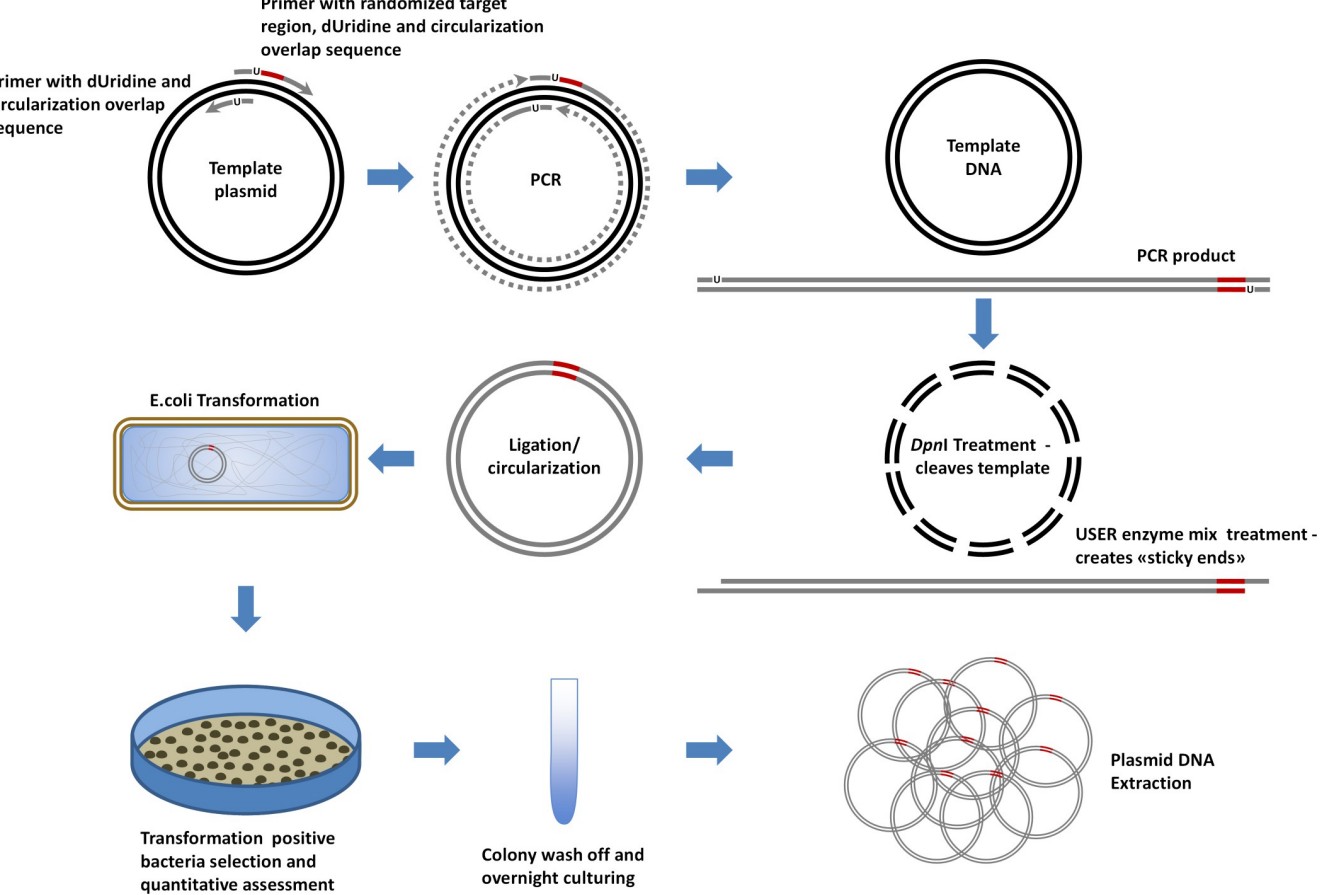

**Fig 1. Randomization was carried out by employing a modified whole plasmid amplification (WPA) strategy, where we utilized 1) one long reverse strand oligonucleotide (starting from the 3' end, it contained a sequence complementary to plasmid DNA, a randomized region (represented by the red line), a stop codon, and additional sequence where the first thymidine was replaced with deoxyuridine (represented by U)), 2) one short forward strand oligonucleotide (starting from the 3' end, it contained a sequence complementary to plasmid DNA and a sequence complementary to the reverse primer's 5' nucleotides (here, also, the first thymidine was replaced with deoxyuridine)), and 3) high fidelity polymerase (Phusion U) which tolerates uridine. Following amplification, the reaction mixture was treated with _Dpn_I restriction enzyme, which cleaves only methylated (bacterial origin) DNA, and USER (Uracil-Specific Excision Reagent) enzyme mix, which excises the uridine base from amplification products and cleaves abasic sites, thus forming a 'sticky end'.** The amplification product was then circularized and transformed into competent _E.coli_ Dh5α strain cells, which were seeded onto an ampicillin-supplemented LB media petri dish and incubated overnight at 37°C. The next day, transformation positive colonies were quantified, washed off, inoculated in ampicillin-containing liquid media, and following overnight culturing, used to extract randomized plasmid DNA. The grey, dotted, and dashed lines represent the synthesized chain, DNA synthesis, and template DNA degradation by _Dpn_I restriction endonuclease, respectively.

**Randomization employing the OverFlap PCR strategy (OverFlapWPA and OverFlapA-symWPA).** The created p426GPD-αfactor-αMSH plasmids (0.2 μg) were cleaved with an _Xho_I restriction enzyme (Thermo Fisher Scientific, Lithuania) in R reaction buffer and 20 μl of the total reaction volume according to the manufacturer's instructions. We verified the reaction's success using agarose gel electrophoresis, and the resultant fragments were purified with a GeneJET PCR Purification Kit. A further 0.06 μg of linearized plasmids were used for WPA as described previously, with two exceptions. First, one of the reactions used asymmetric amplification (Asym) during the initial cycles to avoid exponential amplification of the initial variants (OverFlapAsymWPA). Therefore, we did not add a forward primer, and the total reaction volume was 45 μl. Second, both reactions (OverFlapWPA and OverFlapAsymWPA) were randomized using a Ran-Pept-Lin-Rs primer (Table 1), which, unlike the previously

employed Ran-Pept-Rs primer, contained an additional 26 nucleotide sequence that was complementary to the 3' end of the α-factor secretion signal's CDS, thus covering the 18 nucleotide sequence upstream of the *Xho*I restriction site. Thermal conditions for reaction with both primers (OverFlapWPA) were as described previously. The conditions for OverFlapAsymWPA were as follows: 98˚C for 3 min followed by ten cycles at 98˚C (10 sec), 59˚C (30 sec), and 72˚C (3 min 30 sec) finalized 98˚C (3 min). During the last incubation, 50 pmol of the Ran-Pept-Fw primer was added directly to the reaction mixture, and the reaction was continued for an additional 25 cycles, which were finalized at 72˚C (7 min). We performed all the activities as previously stated. Fig 2 presents the principal scheme of the procedure.

## Massive parallel sequencing employing an IonTorrent personal genome machine (PGM) system

Thus, the randomized region plasmid was amplified using 0.05 μg of plasmid DNA, 1x Phusion U Multiplex PCR Master Mix, 10 pmol of NGS-RanSeq-Fw and NGS-RanSeq-Rs-77 or NGS-RanSeq-Rs-69 primers (Table 1), and water to a final volume of 10 μl. The thermal conditions were as follows: 98˚C for 30 sec followed by 25 cycles at 98˚C (10 sec), 71.5˚C (15 sec),

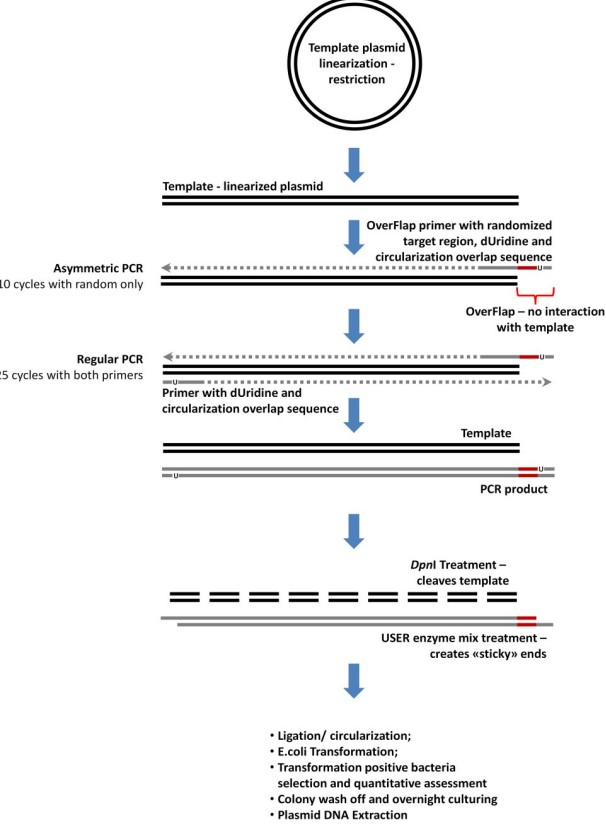

**Fig 2. The principal scheme of the OverFlap strategy for whole plasmid amplification-based randomization of the selected plasmid DNA region.** The methodological approach following circularization is identical to the one described in Fig 1. However, here the plasmid is linearized in a way that the random region of the longer primer does not interact with the template, and an additional stage of asymmetric PCR with only a random region containing a primer is performed to increase the number of sequence variants within the final library. U represents deoxyuridine within the employed primers. The red, black, grey, dotted, and dashed lines represent the randomized region, template DNA, the synthesized chain, DNA synthesis, and template DNA degradation by *Dpn*I restriction endonuclease, respectively.

and 72˚C (15 sec), finalized 72˚C (7 min). Afterward, the repertoire of acquired products was assessed using agarose gel electrophoresis and, in the case of success marked by the presence of a 270 bp fragment, the obtained products were then purified with a NucleoMag NGS Clean-Up and Size Select kit (Macherey-Nagel, Germany) according to the manufacturer's instructions. The quality and acquired amount of the amplicons were assessed using an Agilent High Sensitivity DNA Chip kit on an Agilent 2100 BioAnalyzer (Agilent Technologies, USA) and a Qubit dsDNA HS Assay Kit on a Qubit 2.0 Fluorimeter (Thermo Fisher Scientific, USA).

Each library was diluted to 12 pM and pooled before clonal amplification. We used the Ion PGM™ Hi-Q™ View OT2 kit (Life Technologies, USA) and the Ion OneTouch DL instrument (Life Technologies, USA) for template generation. The template-positive Ion Spheres™ Particles (ISPs) were enriched using Dynabeads MyOne™ Streptavidin C1 beads (Life Technologies, USA) and an Ion OneTouch ES module. ISP enrichment was confirmed using the Qubit 2.0 Fluorimeter (Life Technologies, USA). The sequencing was performed on an Ion 318 v2 chip and Ion Torrent PGM machine employing the Ion PGM™ Hi-Q™ View Sequencing kit (Life Technologies, USA). All the procedures followed the manufacturer's instructions. Each run was expected to produce approximately 150 000 reads per sample. Due to queue and run availability at the genetic analysis facility, sequencing runs for WPA and OverFlapWPA were carried out with a 200 bp target read length. The target read length for OverFlapAsymWPA was 400 bp. After the sequencing procedure, the individual reads were filtered by the PGM software to remove low-quality reads. Sequences matching the PGM 3' adaptor were automatically trimmed. All PGM quality-approved, trimmed, and filtered data were exported as FASTQ files.

### Sequencing data analysis

Following data acquisition, Cutadapt v1.15 was used to test the acquired reads for the presence of sequences that surrounded the randomized region and to extract the randomized region along with the three adjacent nucleotides from both sides (the last codon of the α-factor and stop codon). We further employed the Needleman-Wunsch algorithm implemented in SeqAn 2.4.0, and the acquired reads were aligned against the template (α-MSH plus three adjacent nucleotides from both sides). For additional analyses, we only retained reads for which the first and last trinucleotides matched the template, with lengths divisible by three. The remaining reads were divided into two groups: 1) the target group of insufficiently randomized sequences, which contained reads that aligned with less than ten mismatches and three gaps, and 2) the remaining randomized sequence reads. Since this study initially aimed to create a random peptide expression library, we translated reads from both groups by employing an in-house developed script and quantified all the unique sequences.

### Results

As previous sections explained, the main purpose of this study was to create a plasmid library that would enable the production of random peptides in a yeast *S. cerevisiae* expression system. Since pharmacophores for most biologically active peptides are below 18 aa, we chose to create a plasmid that could secrete peptides (would contain a peptide secretion signal) and would already contain a peptide of similar length whose CDS would subsequently be randomized (due to our previous research experience, we chose α-MSH). We used a p426GPD expression plasmid compatible with the *S. cerevisiae* expression system to produce random peptides. We selected this vector due to its wide application in yeast expression systems, the knowledge that promoter of glyceraldehyde-3-phosphate dehydrogenase (GPD) gene is one of the strongest yeast promoters [60], and recommendations from Dr. Dowell. We employed a PCR-based

mutagenesis approach, which has been successfully used in our laboratory on multiple occasions, to introduce the α-factor secretion signal and α-MSH fusion protein into the selected vector. Therefore, we did not encounter deviations from the initial plan or any unforeseen problems. We created the envisioned plasmid on the first attempt and verified its success by Sanger sequencing.

## Randomization employing a modified WPA strategy

Our initial studies of the literature revealed two suitable strategies for our analysis: cassette mutagenesis, which relies on restriction/ligation for inserting a randomized sequence containing a fragment [51–53], and WPA with a randomized sequence containing an oligonucleotide [6, 50]. Our rather extensive experience in cloning and expression of various mammalian genes suggested that the ligation of two fragments was less efficient than circularization because, unlike the latter, it is a two-stage process. The first stage requires joining two spatially unrestricted DNA ends, while the second stage, in essence, circularizes a newly formed molecule by joining both spatially restricted ends that are located in relative proximity. Therefore, we chose to employ WPA followed by circularization to gain a greater number of clones with a randomized sequence and possibly a higher level of diversity in the randomized sequences. Although we have used this approach in our laboratory on several occasions, because of the recombination-based end joining, we found it to be less reliable than traditional overlapping end PCR-based mutagenesis, which in its final stage involves restriction endonucleases and creates more reliable DNA 'sticky ends'. Thus, we decided to augment the selected strategy with either the introduction of a restriction site after the randomized region or the employment of deoxyuridine-containing primers to create 'sticky ends' after the treatment with USER enzyme mix (contains Uracil DNA glycosylase (UDG) and Endonuclease VIII) [9]. Since the latter creates longer overhangs, that in theory, should increase circularization efficiency, it was the method of choice for our further analyses.

One of the first activities we undertook within the scope of randomization was designing the primers. Our strategy was based on employing one long reverse strand oligonucleotide, which at its 3' end contained an 18 nucleotide-long sequence complementary to the 3' end of the α-factor secretion signal's CDS, followed by 54 random nucleotides, a stop codon, and an additional six nucleotides where the first thymidine was replaced with deoxyuridine. It also contained one short forward strand oligonucleotide, which at its 3' end contained a sequence complementary to 23 reverse strand nucleotides of the p426GPD plasmid immediately following the stop codon of the α-MSH CDS. These were followed by an additional six nucleotides complementary to the reverse primer's 5' nucleotides, and the first thymidine was replaced with deoxyuridine (Table 1).

We selected Phusion U Multiplex PCR Master Mix to amplify the whole plasmid because it is optimized for amplifying difficult targets and, more importantly, it contains high fidelity polymerase (Phusion U) that tolerates the presence of Uridine within the template and growing DNA chains. The amplification was successful, and there was no DNA degradation before or after treatment with USER enzyme mix and *Dpn*I restriction endonucleases, which specifically cleaves only methylated template plasmid DNA (Fig 3). Therefore, the acquired DNA fragment was circularized, transformed into competent cells, and seeded onto a petri dish. The next day, an assessment of the colony forming units by OpenCFU software revealed ~3 802 colonies on our 8.8 mm petri dish (Fig 4A). However, the actual number could have been as much as 20% higher because we observed that the resolution of acquired images (1733×1733 pixels—the maximum for our equipment) was insufficient for the software to reliably identify smaller colonies and distinguish individual ones within dense colony clusters. Yet, OpenCFU

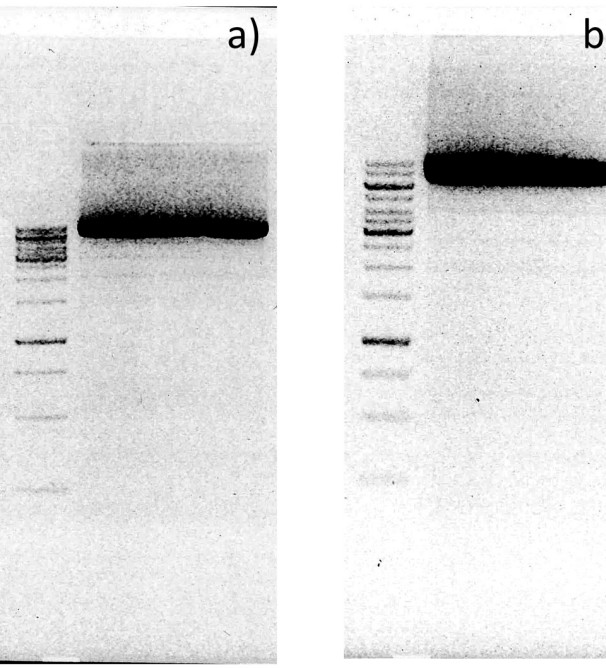

**Fig 3. Visualization of whole plasmid amplification (WPA) products in agarose gel electrophoresis.** a) Prior to the *Dpn*I and USER enzyme mix treatment, and b) after treatment, indicating that the WPA process was successful and the size of the amplification product was not affected by treatment, i.e., no degradation was observed. The first line in each gel contains 2 µl of GeneRuler 1 kb DNA Ladder (Thermo Fisher Scientific, Lithuania), the uppermost band is 10 000 bp, and the three brightest bands are 6 000 bp, 3 000 bp, and 1 000 bp long. The second lane in each gel contains a 5 µl sample of the acquired reaction products.

provided a simple, reliable, and unbiased colony estimation to monitor case-to-case transformation efficiency. Nevertheless, the acquired colony number was lower than expected and insufficient to gain a well-represented random sequence library of 54 nucleotides long.

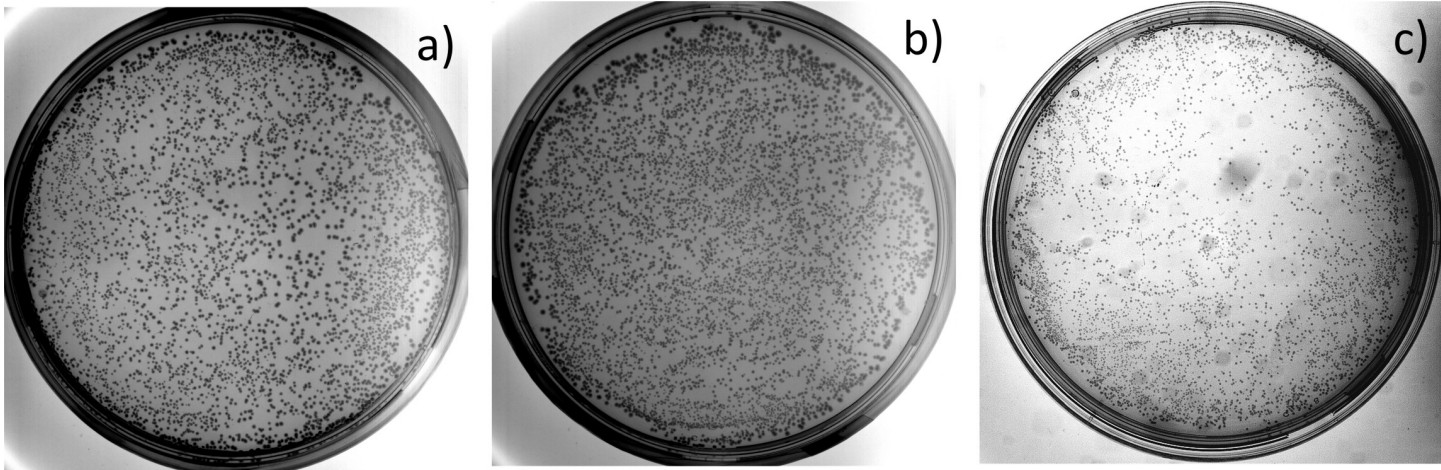

**Fig 4. Photographic pictures of the 8.8 mm diameter petri dish with the selective Amp+ media containing colonies of the *E.coli* Dh5α strain transformed with the circularized product.** a) Whole circular plasmid amplification (WPA) (3 802 colonies), b) OverFlapPCR-based whole plasmid amplification (OverFlapWPA) (4 534 colonies), and c) OverFlapPCR-based asymmetric whole plasmid amplification (OverFlapAsymWPA) (4 865 colonies). All the procedures were completed as described in the Materials and methods section.

However, it was sufficient for assessing randomization process efficiency. Therefore, the colonies were washed off, and the acquired cell culture was inoculated in liquid media for overnight growth and plasmid extraction.

Our initial assessment of randomization process efficiency involved Sanger sequencing. The acquired results confirmed that randomization was successful (Fig 5). However, a major peak could be observed at every position, but they differed between sequenced strands rendering the assessment inconclusive. Therefore, we performed NGS using IonTorrent PGM sequencing technology to comprehensively understand the sequence composition of the created randomized plasmid library.

We prepared the IonTorrent PGM compatible sequencing libraries in a similar manner to the libraries for microbial *16S rRNA* community analysis [61] (i.e., the primers that included both the target sequence and IonTorrent technical sequences (adapters and barcodes) were used to amplify the randomized region). The size of the target amplicon was 273 bp long (vs. 54 bp randomized region). The employed primers were targeted at sequences located 98 bp upstream and 62 bp downstream of the randomized region to provide a reliable reference for identifying the randomized region at the data analysis stage and enable reliable size separation from primer dimers that form during any PCR-based amplification. Since it is a requirement of all NGS technologies, we assessed the quality of the acquired sequencing library using capillary electrophoresis. The results revealed that, contrary to our expectations, products of various sizes formed our library. This indicated that some deletions and duplications occurred (Fig 6 WPA) in addition to the randomization of the target sequence. Subsequently, the acquired sequencing data partially confirmed this suspicion because many randomized translated sequences without a template bias were shorter than 18 aa (54 nucleotides). The data also revealed that ~50% of the acquired reads were highly similar to the template, confirming researchers' concerns that any template-based randomization leads to a considerable bias toward the template sequence, which jeopardizes any attempts for acquiring a random sequence library without a template bias. Additionally, the diversity of the randomized sequences was also lower than expected because 969 unique protein CDSs (vs. ~3 802 colonies) were identified, and only four of these were encoded by ~60% of the reads. Even four of the protein CDSs that resembled the template were encoded by ~49% of the reads. Also, according to acquired data, the average frequency of a specific aa occurrence at every position did not resemble the theoretical frequency of aa occurrence that should be observed in the case of a truly randomized peptide coding library (Tables 2 and S1: sheet 1 and 2). Likewise, the actual number of unique protein CDSs is probably even lower due to the PCR-based library

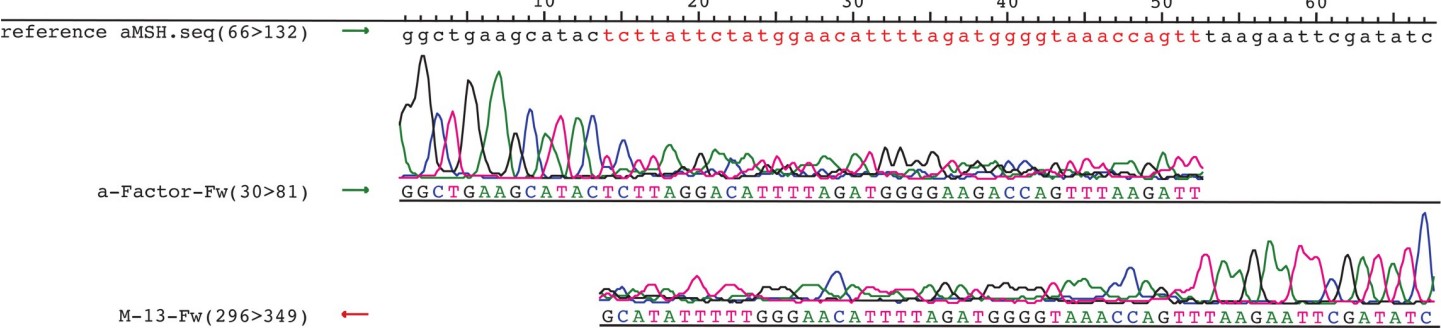

**Fig 5. Assembling whole plasmid amplification (WPA) Sanger sequencing capillary electrophoresis chromatograms.** Sequencing was performed using an a-Factor-Fw primer for forward strand sequencing and an M13-Fw primer for reverse strand sequencing. The fragment containing the sequence template (α-MSH (red font)) was used as a reference for this analysis.

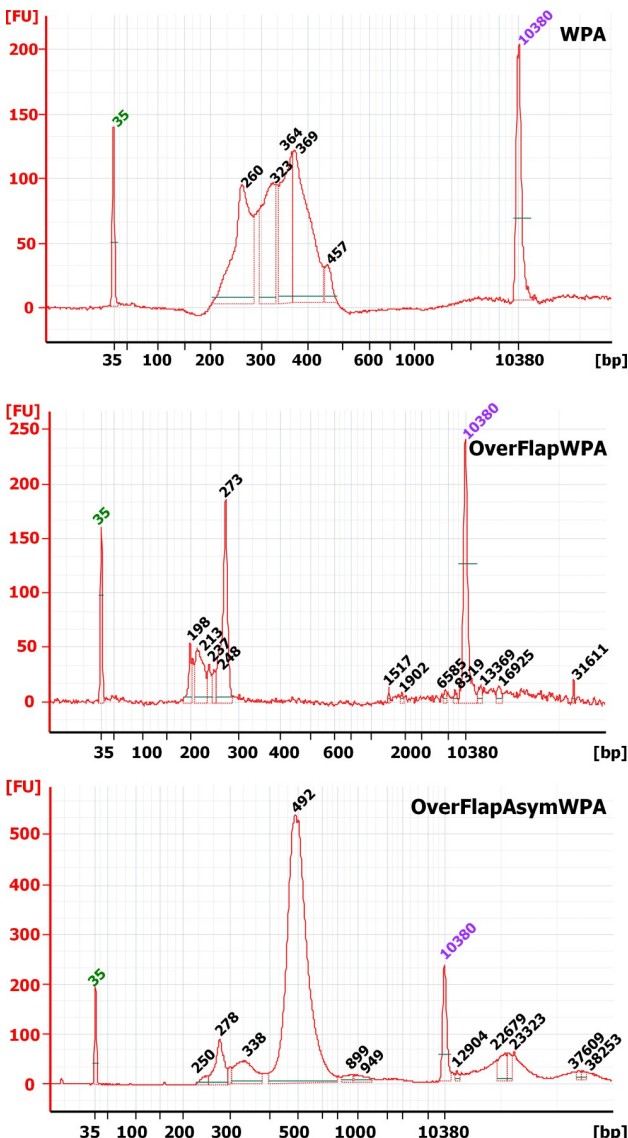

**Fig 6. Chromatograms of the sequencing library fragment sizing analysis carried out on an Agilent Bioanalyzer 2100.** WPA: whole circular plasmid amplification, OverFlapWPA: OverFlapPCR-based whole plasmid amplification, and OverFlapAsymWPA: OverFlapPCR based asymmetric whole plasmid amplification. The 35 bp (green) and 10 380 bp (purple) peaks represent the upper and lower markers employed by data analysis software for the internal calibration of each analysis.

preparation and IonTorrent technology sequencing artefacts. The latter is renowned for its problems with base calling in homopolymers.

Typically, many researchers set a quota for the minimum number of identical reads required for a specific sequence to be considered true to deal with these false reads [53]. However, this approach has several drawbacks. Firstly, it is rather mechanistic and rarely based on careful considerations. Secondly, it can be easily bypassed if a large amount of sequencing data is acquired because the number of false-positive reads increases proportionally. Therefore, we decided to employ an alternative variant pool reconstruction and occurrence probability-based strategy for assessing the acquired variant number.

Hence, we normalized our data by converting it from read counts to fractions of peptide coding reads from the total. Since NGS is a multiple measurement approach that provides a good estimation of the population structure of DNA sequences within a library, acquired fractions can also be considered as probabilities for detecting a specific peptide coding variant if we performed Sanger sequencing on a single colony's content. This is true because most colonies contain a single type of plasmid and double transformations are rare [62]. We can also extrapolate that the total number of transformation-positive colonies reflects the total number of variants within our library and, subsequently, the total amount of single colony Sanger sequencing required to cover the whole pool of variants. Therefore, by multiplying the fraction of a specific variant with the total number of variants (i.e., colony count), we acquired an approximate reconstruction of the whole library variant pool (i.e., how many colonies contained which variant). In an ideal situation, all the numbers acquired in such a way should belong to the natural number space. However, due to measurement errors induced by the whole NGS workflow, differences in colony sizes, unequal plasmid copy numbers within bacterial cells, and many other reasons, ours were fractions. Although numbers equal to or above one provided a good indication that a specific variant was indeed present in our library, those below one were more akin to presence probabilities. The estimation results were treated as reconstructions and probabilities, whereas different types of DNA sequence reads, which encoded identical peptides, were summed. Hence, we named them 'cumulative probabilities of occurrence' (CPO).

Consequently, we divided the whole variant repertoire into two groups: those with a CPO value equal to or above one and those with a CPO below one. Then, we calculated the CPO value of each group by summing the CPO values of each variant, thus acquiring a rough assessment of the number of colonies that, in theory, would have been occupied by members of each group. We calculated the number of variants present in our library by summing the variant count within the first group with the CPO value of the second group, rounded to the nearest whole number. The latest corresponds to the approximate number of colonies occupied by members of this group. Likewise, since most bacterial cells are transformed by a single variant, it was closer to the actual number of variants in the second group.

After applying these calculations, we concluded that the WPA approach created only 185 unique protein CDSs (Table 2). Although this approach is complicated, significantly affected by colony estimation accuracy, and has differences in replication speed for various plasmid variants during bacterial culturing in liquid media, it has several advantages. First, it can be calculated and individually applied to each data set. Second, it is less sensitive to fluctuations in sequencing data amount as it is frequency-based.

## Randomization employing the OverFlapPCR-based WPA strategy (OverFlapWPA)

As mentioned previously, the only proposed solutions for mitigating the template bias problem were adjusting the annealing temperature and loop-out/loop-in excision of the target sequence from the template, but since we intended to randomize 54 bp region the temperature adjustment was not a feasible option, and excision of the α-MSH CDS would simply switch the template to a sequence that is located downstream of the α-MSH CDS. We believed that none of these approaches would solve the issue. Therefore, we devised a novel strategy to linearize the prior PCR-based WPA template at the position located a short distance upstream, or a short distance downstream of the intended randomization site, ensuring that the template ended with the randomization primer's target site (the part that was perfectly complementary to the template). During WPA, the randomized part would form an overhang that 'freely

**Table 2. General characteristics of the acquired sequencing data and an overview of the results of the subsequent data analysis.** WPA represents randomization through whole circular plasmid amplification, OverFlapWPA represents OverFlapPCR based whole plasmid amplification, and OverFlapAsymWPA represents Over-FlapPCR based asymmetric whole plasmid amplification.

| | WPA | OverFlapWPA | OverFlapAsymWPA |
|---|---|---|---|
| **Number of observed colonies** | **3 802** | **4 534** | **4 865** |
| **Number of Sequencing reads** | **140 960** | **139 461** | **806 671** |
| **Mean Read Length** | **147 bp** | **141 bp** | **163 bp** |
| **Number of target region sequence reads (passed quality filters)** | **62 636 (44.44%)** | **94 609 (67.84%)** | **611 478 (75.8%)** |
| *[1]Number of reads that encode peptides with CPO ≥ 1* | *60 977* | *87 791* | *103 747* |
| *Number of reads that encode peptides with CPO <1* | *1 659* | *6 818* | *507 731* |
| **Number of unique protein coding sequences** | **969** | **4 636** | **51 504** |
| *[1]Number of unique protein coding sequences with CPO ≥ 1* | *84* | *464* | *627* |
| *Number of unique protein coding sequences with CPO <1* | *885* | *4 172* | *50 877* |
| **CPO of all unique protein coding sequences** | **3 802.00** | **4 534.00** | **4 865.00** |
| *CPO of all unique protein coding sequences with individual CPO ≥ 1* | *3 701.30* | *4 207.25* | *825.42* |
| *[2]CPO of all unique protein coding sequences with individual CPO <1* | *100.70* | *326.75* | *4 039.58* |
| **Number of insufficiently randomized reads (% of total)** | **31 447 (50.21%)** | **3 (0.003%)** | **0 (0%)** |
| *Number of insufficiently randomized reads that encode peptides with CPO ≥ 1* | *31 110* | *0* | *0* |
| *Number of insufficiently randomized reads that encode peptides with CPO <1* | *337* | *3* | *0* |
| **Number of unique insufficiently randomized protein coding sequences** | **117** | **1** | **0** |
| *[3]Number of unique insufficiently randomized protein coding sequences with CPO ≥ 1* | *9* | *0* | *0* |
| *Number of unique insufficiently randomized protein coding sequences with CPO <1* | *108* | *1* | *0* |
| **CPO of all unique insufficiently randomized protein coding sequences** | **1 908.83** | **0.15** | **0.00** |
| *CPO of all unique insufficiently randomized protein coding sequences with individual CPO ≥ 1* | *1 888.37* | *0.00* | *0.00* |
| *[4]CPO of all unique insufficiently randomized protein coding sequences with individual CPO <1* | *20.46* | *0.15* | *0.00* |
| **Number of randomized reads (% of total)** | **31 189 (49.79%)** | **94 606 (99.997%)** | **611 478 (100%)** |
| *Number of randomized reads that encode peptides with CPO ≥ 1* | *29 867* | *87 791* | *103 747* |
| *Number of randomized reads that encode peptides with CPO <1* | *1 322* | *6 815* | *507 731* |
| **Number of unique randomized protein coding sequences** | **852** | **4 635** | **51 504** |
| *[5]Number of unique randomized protein coding sequences with CPO ≥ 1* | *75* | *464* | *627* |
| *Number of unique randomized protein coding sequences with CPO <1* | *777* | *4 171* | *50 877* |
| **CPO of all unique randomized protein coding sequences** | **1 893.17** | **4 533.86** | **4 865.00** |
| *CPO of all unique randomized protein coding sequences with individual CPO ≥ 1* | *1 812.92* | *4 207.26* | *825.42* |
| *[6]CPO of all unique randomized protein coding sequences with individual CPO <1* | *80.25* | *326.60* | *4 039.58* |
| **[1+2] Number of unique protein coding sequences within libraries** | **185** | **791** | **4667** |
| *[3+4] Number of unique insufficiently randomized protein coding sequences within libraries (% of total colonies)* | *30 (50.21%)* | *0 (0%)* | *0 (0%)* |
| *[5+6] Number of unique randomized protein coding sequences within libraries (% of total colonies)* | *155 (49.79%)* | *791 (100%)* | *4667 (100%)* |

* Expected minimal abundancy in %, calculated as 100% divided by number of observed colonies on a petri dish

flapped' over the end of the template (Fig 2), having minimal interaction with it and mitigating the bias effect. Hence, we named this approach OverFlap PCR or OverFlap WPA.

In our case, we were able to identify the restriction site that was conveniently located 21 bp upstream of the α-MSH CDS within the α- secretion factor CDS. All other processes were carried out identically to WPA and the OpenCFU estimated colony number was 4 534 (Fig 4B). Although a little higher, the number was still lower than necessary for acquiring a well-represented random sequence library but it was sufficient to evaluate its diversity. Sanger sequencing of the acquired plasmid library confirmed that randomization, per se, had happened.

Therefore, we proceeded with generating a sequencing library and assessing its quality. Data from BioAnalyzer revealed that the proportion of shorter and longer fragments was significantly decreased, and the single peak of the 273 bp target could be identified (Fig 6 OverFlapWPA). NGS analysis revealed that, compared to traditional WPA, the number of unique protein CDSs increased more than four times, reaching 4 636. Overwhelmingly, most reads (99.997%) were randomized, bearing little or no similarity to the α-MSH CDS. In addition, the abundance of the most represented sequence was below 5% and the total abundance of the four most represented sequences decreased to only ~8.2%. Also, although applying the previously presented CPO-based calculations significantly reduced the number of unique reads to 791, it was still more than four times higher than that of WPA. Similarly, the average frequency of specific aa occurrence at every position displayed a greater similarity to the theoretical frequency of aa occurrence in a truly randomized peptide coding library, highlighting that significant improvements were achieved (Tables 2 and S1: sheet 1 and 2). Collectively, these results demonstrated that sequence diversity within OverFlapWPA-generated libraries was significantly greater than within those generated by the traditional WPA approach.

## Randomization employing an asymmetric OverFlapPCR-based WPA strategy (OverFlapAsymWPA)

Despite the previously described success, we believed we could still improve the sequence diversity within acquired libraries because ~8.2% of reads were encoding the four most represented sequences. We considered that this phenomenon might have been related to PCR peculiarities, in which sequences generated during the first cycles are amplified faster than others. In our view, there were two solutions to this problem. Firstly, increase the template concentration and decrease the amplification cycles. Secondly, perform several cycles of asymmetric PCR with a randomization primer to linearly increase the concentration of already randomized templates without introducing an early cycle amplification bias. Since employing the first option might increase the template background within the randomized library, we decided to proceed with the second. Thus, we carried out the first ten cycles of the PCR procedure using only a randomization primer, while the remainder of the process was identical to the previous ones. These activities generated a plasmid library acquired from 4 865 transformation-positive colonies (Fig 4C). Again, we used Sanger sequencing to confirm the success of the randomization procedure. However, the agarose gel electrophoresis and capillary electrophoresis-based DNA fragment analysis results were peculiar. They revealed that the size of the library's major PCR product was 492 bp, while the amount of expected 273 bp fragments, although shifted by 5 bp, was significantly lower (Fig 6 OverFlapAsymWPA). Therefore, we decided to perform an additional purification with size selection beads to remove any DNA fragments larger than 300 bp. However, analysing the acquired DNA sample revealed the same pattern—major peak was larger than expected. Further attempts with ~250 bp band excision from agarose followed by DNA purification returned similar results. Therefore, we concluded that the larger fragments were 'products' of our target fragment. Our best speculation was that due to the high level of randomization and PCR-related cyclic temperature-induced denaturation-renaturation, there was a significant proportion of double-stranded DNA molecules within our sequencing library for which both strands were not perfectly matched. This left these unmatched bases free to interact with other DNA molecules and form something akin to G-quadruplexes [63]. However, since these interactions were due to the random nature of the library sequences and not the result of intelligent design or evolutionary selection, they were unstable. It is possible that these quadruplexes were constantly forming and disbanding during DNA travel through the electrophoresis polymer, potentially shifting the peak/band size towards something smaller

than the sum of both molecules. However, our numerous purifications resulted in a significant loss of the sequencing library. Therefore, it was recreated, purified, and sequenced as described in the Materials and methods section. The acquired sequencing data also confirmed that the input library was of the correct size because despite the coincidental sequencing of 400 bp long reads (performed due to requirements of other libraries sequenced during a specific sequencing run), the mean read length was only 163 bp. Likewise, this sequencing run resulted in a significantly larger amount of data (806 671 reads) than the previous runs (~140 000 reads). Thus, all the abundance and diversity results should be considered in this context. The sequencing data analysis revealed that, compared to OverFlapWPA, the number of unique protein CDSs was 11 times higher, reaching 51 504, which is a considerable increase resulting from a data amount that was only ~6.5 times larger. Interestingly in spite of this increased data amount, all of the reads were randomized, bearing little or no similarity to the α-MSH CDS. Moreover, the abundance of the most represented sequence was below 0.1%, and the total abundance of the four most represented sequences was below 0.25%. Although applying CPO-based calculations significantly decreased the number of unique variants to 4 667, it was still almost six times higher than that of OverFlapWPA. Similarly, we observed further improvements when calculating the average frequency of specific aa occurrence at every position. Here, experimentally acquired numbers greatly resembled the theoretical frequencies of aa occurrence within a truly randomized peptide coding library (Tables 2 and S1: sheet 1 and 2). Thus, it was clear that introducing an asymmetric PCR stage resulted in a random plasmid library with even greater sequence diversity.

## Discussion

Recent developments in process automatization and high throughout data acquisition technologies have started a new era in many life and medical science research fields. Protein engineering and synthetic biology have greatly benefited from introducing these technologies into their everyday practices by being at the frontline of knowledge and trying to develop something that cannot be found or has not been observed in nature through a series of creation and effect observation experiments. Introducing NGS technologies has been particularly beneficial to disciplines that aim to study or improve proteins through CDS randomization. Although there are multiple methodologies for randomizing target sequences that fit various research strategies, the current technological challenge involves reliably randomizing protein segments larger than 9 bp. A strategy including cassette replacement requires the convenient localization of two restriction sites and, due to the inefficiency of two-fragment ligation, produces a relatively low number of transformation-positive colonies. However, as demonstrated in this study, PCR-based WPA suffers from significant randomization bias towards the template sequence.

The OverFlap PCR strategy that we present in this study provides a simple and reliable solution to said sequence bias problem. As our OverFlapWPA experiment demonstrated, linearizing the plasmid at the position adjacent to the randomization site prevents the primer's random sequence-containing segment from interacting with the template during amplification, resulting in an unbiased sequence library. Moreover, our OverFlapAsymWPA experiments demonstrated that even higher sequence diversity can be achieved if asymmetric PCR principles are applied during the first cycles of OverFlapWPA. An additional problem that we encountered during our initial WPA experiments was the high prevalence of reads that were shorter than the randomized region (Fig 6 WPA, S1 Table: sheet 1). Although we cannot provide any reasonable explanation for this phenomenon beyond mismatch-induced degradation by either polymerase or *E.coli* repair machinery, it seems that introducing asymmetric amplification resolved the issue (Fig 7).

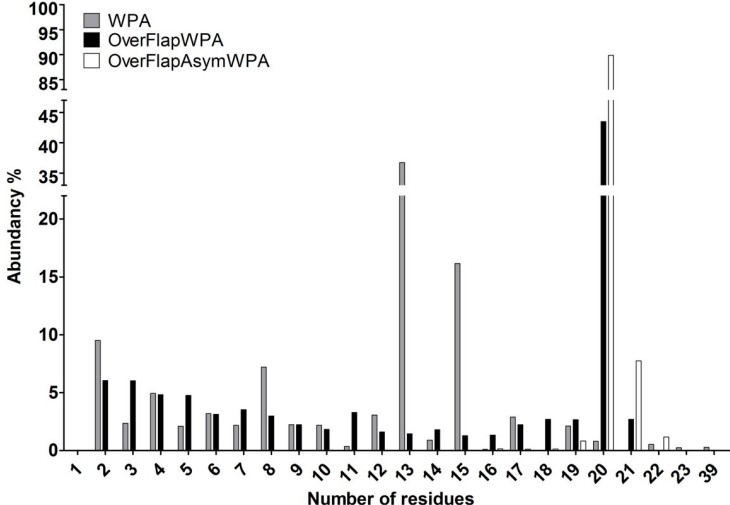

**Fig 7. Quantification of randomized region encoded peptides according to size.** As the Materials and methods section describe, in addition to randomized region the data also includes the last aa of the α-factor secretion signal CDS and stop codon. In the case of '1', last aa of α-factor secretion signal CDS was missing.

Although it is very effective for acquiring a randomized library without a template bias, the described methodology has several drawbacks.

Firstly, the number of acquired transformation-positive colonies is far too low to ensure that all the possible variants are represented in an acquired library of larger randomization. According to very rough calculations, on average, we were able to acquire ~4 400 colonies per transformation. Thus, under ideal conditions, where all possible sequence variants are evenly represented, this number shall be sufficient to reliably randomize no more than two codons ($4^6$ = 4 096). Of course, this situation would be less of a problem for researchers interested in identifying novel biologically active peptides because having a library of 4 000 reliably random peptides is better than having a library of 4 000 peptides with a significant bias towards the template. However, for those wanting to randomize a specific section of a known protein during a single experiment, this might be a significant drawback. In our opinion, there are two solutions to this problem. The first would involve performing repeated randomizations and transformation-positive colony collections until the desired number of colonies or the level of variations determined by NGS is reached. The second solution necessitates employing an entirely different circularization and transformation strategy. We believe the best alternative would utilize Gibson assembly coupled with electroporation. According to several authors, employing this approach allows up to $10^8$ colony forming units to be acquired [49]. An additional advantage of Gibson assembly would be the correction of possible mismatches between strands because, due to its proximity to the end of the DNA fragment, the randomized region for one of the strands shall be destroyed and rebuilt anew using the remaining strand as a template.

Secondly, relying on restriction for linearizing the template limits the choice of randomization sites. The workarounds, in this case, would be either introducing restriction sites within the template through site-directed mutagenesis or linearization through WPA. Introducing a restriction site in the first case would matter little as long as it is unique within the chosen template because, during the randomization process, altered nucleotides can be reverted to the desired state. The advantage of the restriction enzyme approach is that after randomization, the *Dpn*I restriction enzyme can be used for reliable fragmentation/destruction of template DNA. However, the disadvantage is the need to perform additional mutagenesis. Conversely,

WPA is advantageous because the plasmid can be linearized at virtually any site within a single step. However, due to a lack of methylation, such a template would be chemically indistinguishable from the newly created strands, and selective digestion by the *Dpn*I restriction enzyme would not be possible. Nevertheless, we believe that due to its simplicity, the WPA approach would be more attractive than introducing a restriction site, and we currently see two workarounds for mitigating the template background issue. The simplest one would exclude the intended overlap and randomize target regions from the linearized template by employing PCR primers that anneal to flanking regions of the randomization target and are directed away from it. Thus, the lack of a complementary region would significantly decrease the probability of circularization, while in the case of Gibson assembly, end-joining shall be nearly impossible. The other workaround would be to use a modified dNTP mix, where thymidine triphosphate (dTTP) is replaced with deoxyuridine triphosphate (dUTP) in combination with uridine tolerant high fidelity polymerase for template amplification (such as dNTP/dUTP Mix and Phusion U Hot Start DNA Polymerase (Thermo Fisher Scientific, USA)). In this case, adding the previously described USER enzyme mix to the post-randomization mixture would result in cleavage and elimination of the created linear DNA template, similar to *Dpn*I's cleavage of methylated sites. Also, both mentioned workarounds are not mutually exclusive and can be employed in combination.

We created an additional methodological workflow, available in the Supplementary material section (S2 Fig), to collectively address the potential improvements to our randomization methodology pipeline.

Although the methodology presented here concerns a WPA strategy, other randomization strategies can consider not including a template at the randomization site. For example, any probe-based randomization should consider using a fragmented template with gaps at selected randomization sites instead of a single and large template. Additionally, as Gibson assembly can be applied to create a single construct from multiple fragments [64], our OverFlap PCR approach can also be employed for randomizing multiple sites.

In summary, we present a novel approach for reliably introducing random and unbiased nucleotide sequences into virtually any site of a plasmid, which we named "OverFlap PCR" to distinguish it from "Overhang PCR" and "Overlap PCR". The method was based on employing a randomized region containing a primer and a linearized plasmid template in partially asymmetric WPA followed by circularization. Plasmid linearization was carried out to prevent or minimize the primer's randomized region from interacting with the template. Massive parallel sequencing on an IonTorrent PGM machine confirmed that the acquired library was random and displayed no template bias. At the end of the article, we also discussed the drawbacks of the developed methodology and presented a plan for future improvements.

## Supporting information

**S1 Fig. The feature map of the created p426GPD-aFactor-aMSH plasmid that this study used for generating the plasmid library for random peptide expression.** Creating the p426GPD-aFactor-aMSH plasmid involved supplementing the standard p426GPD vector with a *Saccharomyces cerevisiae* yeast α-factor secretion signal and the α-melanocyte stimulating hormone fusion protein-coding sequence. *Bam*HI and *Eco*RI restriction sites were used for this purpose. This study used an *Xho*I restriction site for plasmid linearization before OverFlap PCR.
(PDF)

**S2 Fig. The principal scheme of the proposed OverFlap whole plasmid amplification-based randomization of selected plasmid DNA regions.** The red, black, grey, and dotted lines

represent the randomized region, template DNA, synthesized chain, and DNA synthesis, respectively.
(PDF)

**S1 Table. List of peptide variants that were identified and the average frequency of a specific aa occurrence at every position.** Sheet 1: list of variants that were identified during each experiment, number of reads that encode each peptide, and their relative abundance within each dataset and CPO. Variants with CPOs <1 are shaded in red. Sheet 2: average frequency of a specific aa occurrence at every position during each experiment and comparison to their theoretical occurrence in random nucleic acid sequences.
(XLSX)

**S1 Raw images.**
(PDF)

## Acknowledgments

We would like to thank Dr. Simon Dowell for providing the necessary plasmids and aiding the selection of plasmids with appropriate promoters and Ms. Linda Lazdiņa for her input in modifying the source plasmids.

We would like to thank M.Sc. Ņikita Zrelovs for fruitful discussions on various options of data presentation.

## Author Contributions

**Conceptualization:** Ance Roga, Davids Fridmanis.

**Funding acquisition:** Davids Fridmanis.

**Investigation:** Artis Linars, Ivars Silamikelis, Dita Gudra, Ance Roga, Davids Fridmanis.

**Resources:** Dita Gudra.

**Software:** Ivars Silamikelis.

**Supervision:** Davids Fridmanis.

**Writing – original draft:** Davids Fridmanis.

**Writing – review & editing:** Ivars Silamikelis, Dita Gudra, Ance Roga, Davids Fridmanis.

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
