## [Decision Letter · Decision Letter 0]

10 Feb 2022

PONE-D-21-40826OverFlap PCR – a reliable approach for generation of plasmid DNA libraries containing random sequences without template bias.PLOS ONE

Dear Dr. Dāvids Fridmanis,

Thank you for submitting your manuscript to PLOS ONE. After careful consideration, we feel that it has merit but does not fully meet PLOS ONE’s publication criteria as it currently stands. Therefore, we invite you to submit a revised version of the manuscript that addresses the points raised during the review process.

ACADEMIC EDITOR: Please find the reviewers comments below

We look forward to receiving your revised manuscript.

Kind regards,

Shawky M Aboelhadid, PhD

Academic Editor

PLOS ONE

Journal Requirements:

3.Thank you for stating the following in the Funding Section of your manuscript: 

"This work was supported by European Regional Development Fund (ERDF) [Project No.: 1.1.1.1/16/A/055]; "

"AL was supported by European Regional Development Fund (ERDF) [Project No.: 1.1.1.1/16/A/055];

AR was supported by European Regional Development Fund (ERDF) [Project No.: 1.1.1.1/16/A/055];

DF was supported by European Regional Development Fund (ERDF) [Project No.: 1.1.1.1/16/A/055];

Reviewers' comments:

Reviewer's Responses to Questions

**Comments to the Author**

1. Is the manuscript technically sound, and do the data support the conclusions?

Reviewer #1: Yes

Reviewer #2: Yes

2. Has the statistical analysis been performed appropriately and rigorously? 

Reviewer #1: Yes

Reviewer #2: I Don't Know

3. Have the authors made all data underlying the findings in their manuscript fully available?

Reviewer #1: Yes

Reviewer #2: Yes

4. Is the manuscript presented in an intelligible fashion and written in standard English?

Reviewer #1: Yes

Reviewer #2: No

5. Review Comments to the Author

Reviewer #1: 1. The Introduction meanders, as it reviews a broad range of methods, some tangential to this study, without offering much justification for those they developed. It’s not obvious that an unbiased library of 18 amino acid peptides would contain more functional clones than simpler libraries created with existing methods. Once that argument has been articulated, the authors can explain why existing methods are inadequate.

2. The authors have an epistemological problem. Older methods were analyzed by low throughput Sanger sequencing, whereas they applied higher throughput Next Generation Sequencing. So we cannot conclude that their methods produce less biased libraries unless they were to reproduce an older method and analyze the resulting library via NGS sequencing.

3. What fraction of 18 amino acid sequence space is viable in E. coli AND S. cerevisiae? The authors focus exclusively on the biases of in vitro library construction, but it is quite possible that the biases (and limitations in transformation efficiency) imposed by host cells will preclude the discovery of functional peptides.

4. Moreover, methods that link mRNA to their translation products in vitro (via ribosome display, encapsulation of mRNAs with working ribosomes, or a cross-linking puromycin analog) have produced large and diverse libraries. Those methods, however, have not been very widely adopted, in part because they are technically challenging but also because the results haven’t been as impressive as those produce by in silico design of de novo proteins. So bias in existing library construction methods might not be the bottleneck that the authors believe it to be.

5. Others have shown that the cloning efficiency of whole plasmid amplification products is inversely proportional to length. So the methods described here cannot be adapted to every application.

6. The numbers of colonies analyzed (<5000, Table 2) was miniscule compared to the number of possible sequences (4^54) or the number of NGS reads. Why produce such diverse libraries (or apply such high throughput analytic techniques) if transformation efficiency imposes such a tight bottleneck? The authors believe that the problem can be overcome via Gibson Assembly but it is probable that the transformation protocol (heat shock, growth of colonies on agar plates) is limiting.

7. The calculation of cumulative occurrence probability (COP) ought to be spelled out so that all readers (including this reviewer) can understand it.

Reviewer #2: The manuscript "OverFlap PCR-a reliable approach for generation of plasmid DNA libraries containing random sequence without template bias" describes a straightforward method for generating randomized protein sequences without template bias, which is common and problematic with current PRC based methods. It's clear that the authors are experts in PCR techniques and have a strong understanding of the current issues with template bias in the pursuit of obtaining fully randomized sequences and the importance of this to identifying new therapeutic candidates. I feel this is an important paper that could have a significant impact in the quest to obtain fully randomized sequence with a relatively simple approach.

1. The manuscript is technically sounds with data supporting their conclusions. There are a few comments I would like to have addressed.

a. Figure 2 is a little confusing. The figures focus is asymmetric PCR, but the schematic shows both primers binding to the template. Perhaps, first show the overFlap primer binding to the template by itself, then below that, show the template again but this time with the addition of the sequence primer.

b. the labeling of the primers is confusing. Based on figure 1 (circular plasmid) and Figure 2, it appears to me that the "overFlap" primer is the forward primer, but in the texts and in table 1 its called the reverse primer. What appears to be the reverse primer in the the figure is called the forward primer. Perhaps this is based on the primer location shown on the linearized plasmid in figure 1. However, based on all the other diagrams and the bulk of this work...this labeling is confusing.

c. In Figure 4, its reported that plate C has the most colonies, however, visually, it doesn't appear to me to be the case. It appears to have the fewest colonies.

2. The authors appear to have expertise on the mathematical calculations. However, I had a hard time following their math/logic, especially as it relates to the explanation of their approach to calculating cumulative occurrence probability (COP). For example, how did you arrive at the value for "probability of each variant occurrence"? The description related to COP needs to be more more clear.

3. It appears the authors have made available all their data related to this work.

4. This manuscripts need a lot of work with the writing. My biggest issue with the paper is that it is not clearly written. It has many typos, dropped words, and poorly constructed sentences. I recommend having this manuscript thoroughly edited by someone who speaks native English or at least has a stronger command of the English written language. There are too many typos and grammatical errors for me to address individually. However, I do want to highlight a couple issues.

a. Page 12 under heading "Randomization employing modified...", second paragraph you wrote a couple times that "9 nucleotides where the first thymidine was replaced deoxy uridine". However, in Table 1 (and on page 8), you show 6 nucleotides were added prior to the deoxy uridine.

b. Figure 8 and its detailed explanation is awkward and out of place in the discussion. It also seem a bit redundant to the earlier figures. I do think its very useful information. I suggest you consider moving this to supplemental data.

6. PLOS authors have the option to publish the peer review history of their article (what does this mean?). If published, this will include your full peer review and any attached files.

Reviewer #1: No

Reviewer #2: No

---

## [Author Response · Author response to Decision Letter 0]

10 May 2022

Responses to Journal Editing Team Comments:

 Comment: 1. Please, ensure that your manuscript meets PLOS ONE's style requirements, including those for file naming. The PLOS ONE style templates can be found at 

 Response: We thank Editors for the links to templates. We have accordingly updated the style of the manuscript.

 Comment: 2. We note that the grant information you provided in the ‘Funding Information’ and ‘Financial Disclosure’ sections do not match. When you resubmit, please ensure that you provide the correct grant numbers for the awards you received for your study in the ‘Funding Information’ section.

 Response: Thank You for following the consistency of provided funding source information. We carefully checked the grant information in both sections and, unfortunately, we were unable to identify any discrepancies in grant numbers. The only differences were indications of specific authors that were supported by specific funding source. However this issue was resolved through deletion of ‘Funding Information’ section from the Manuscript.

 Comment: 3.Thank you for stating the following in the Funding Section of your manuscript: "This work was supported by European Regional Development Fund (ERDF) [Project No.: 1.1.1.1/16/A/055];" We note that you have provided funding information that is not currently declared in your Funding Statement. However, funding information should not appear in the Acknowledgments section or other areas of your manuscript. We will only publish funding information present in the Funding Statement section of the online submission form. 

"AL was supported by European Regional Development Fund (ERDF) [Project No.: 1.1.1.1/16/A/055];

AR was supported by European Regional Development Fund (ERDF) [Project No.: 1.1.1.1/16/A/055];

DF was supported by European Regional Development Fund (ERDF) [Project No.: 1.1.1.1/16/A/055];

 Response: The information within “Funding Statement” was created following provided PLOS ONE guidelines. However, although some of the authors did not receive remuneration from indicated funding source, all other expenses were covered by it, thus we would prefer that following “Funding Statement” is included “This work was supported by European Regional Development Fund (ERDF) [Project No.: 1.1.1.1/16/A/055]”

 Comment: 4. PLOS ONE now requires that authors provide the original uncropped and unadjusted images underlying all blot or gel results reported in a submission’s figures or Supporting Information files. This policy and the journal’s other requirements for blot/gel reporting and figure preparation are described in detail at https://journals.plos.org/plosone/s/figures#loc-blot-and-gel-reporting-requirements and https://journals.plos.org/plosone/s/figures#loc-preparing-figures-from-image-files. When you submit your revised manuscript, please ensure that your figures adhere fully to these guidelines and provide the original underlying images for all blot or gel data reported in your submission. See the following link for instructions on providing the original image data: https://journals.plos.org/plosone/s/figures#loc-original-images-for-blots-and-gels. 

 Response: Thank You for noting this shortcoming in the original submission. We have created an additional supplementary file “S4_raw_images.pdf”, which contains raw gel images.

 Comment: 5. We note that you have included the phrase “data not shown” in your manuscript. Unfortunately, this does not meet our data sharing requirements. PLOS does not permit references to inaccessible data. We require that authors provide all relevant data within the paper, Supporting Information files, or in an acceptable, public repository. Please add a citation to support this phrase or upload the data that corresponds with these findings to a stable repository (such as Figshare or Dryad) and provide and URLs, DOIs, or accession numbers that may be used to access these data. Or, if the data are not a core part of the research being presented in your study, we ask that you remove the phrase that refers to these data.

 Response: Thank You for addressing this issue. Since the Sanger sequencing was used as the means for preliminary verification of randomization process, which triggered implementation of further activities, while NGS provided all the main results, we don’t consider former as sufficiently relevant for presentation in this article, therefore we have removed reference to unpresented data. 

Reviewer's Responses to Questions

 Comment: 1. Is the manuscript technically sound, and do the data support the conclusions?

Reviewer #1: Yes

Reviewer #2: Yes

 Response: We thank both reviewers for positive assessment on this point;

 Comment: 2. Has the statistical analysis been performed appropriately and rigorously? 

Reviewer #1: Yes

Reviewer #2: I Don't Know

 Response: We thank reviewer for their input in assessment of this point;

 Comment: 3. Have the authors made all data underlying the findings in their manuscript fully available?

Reviewer #1: Yes

Reviewer #2: Yes

 Response: We did our best to provide all data that was underlying our research and we thank both reviewers for their positive response;

 Comment: 4. Is the manuscript presented in an intelligible fashion and written in standard English?

Reviewer #1: Yes

Reviewer #2: No

 Response: We thank reviewers for reading our manuscript. Professional English editing has been carried out to improve its readability.

Responses to Reviewer #1 Comments:

 Comment: 1. The Introduction meanders, as it reviews a broad range of methods, some tangential to this study, without offering much justification for those they developed. It’s not obvious that an unbiased library of 18 amino acid peptides would contain more functional clones than simpler libraries created with existing methods. Once that argument has been articulated, the authors can explain why existing methods are inadequate.

 Response: We thank reviewer for sharing his/hers thoughts on the subject. However we have to respectfully disagree in majority of points that have been voiced in this comment. In our opinion development of novel methodology, should always be presented in context with exist sting methods, unless it is unique and defines a novel research direction. The introduction section of our manuscript was fashioned to resemble our journey through scientific literature that we carried out while developing presented methodological approach. The advantages and drawbacks of all mentioned mutagenesis/randomization method groups are integrated into narrative and usually located at the end of respective paragraph. We believe that sufficient amount of argumentation has been presented to justify the development of presented approach. Also adequacy or inadequacy of existing method is a matter of goal and opinion, thus we don’t consider any of existing methods as generally inadequate and wouldn’t place such claims in our manuscript. However most of them indeed did not suit our future needs and presented research clearly demonstrates that template bias greatly decreases the number of functional variants and often results in quantitative dominance of only several. However we agree with the reviewer that consequences of bias problem are not sufficiently emphasized in the introduction section. Therefore we provided an elaboration on this issue.

 Comment: 2. The authors have an epistemological problem. Older methods were analyzed by low throughput Sanger sequencing, whereas they applied higher throughput Next Generation Sequencing. So we cannot conclude that their methods produce less biased libraries unless they were to reproduce an older method and analyze the resulting library via NGS sequencing.

 Response: We thank Reviewer for the valuable comment. It is true that predominant majority of previous studies did not employ NGS for assessment of randomization efficiency and we agree that conclusions on randomization efficiency should not be drawn unless one of the older methods is reproduced and acquired results are compared to those of novel approach. However, it should be noted that our manuscript describes the pipeline of experimental procedures and majority of these are well known and widely used in many laboratories worldwide. Thus the deoxy uridine - USER enzyme (Uracil-Specific Excision Reagent) based cloning was described by Bitinaite et al. in 2007 [9], while employed procedure for chemical transformation of competent E.coli cells was published by Green & Rogers in 2013 [58]. Therefore the evaluation of individual procedure efficiency is not the goal of this research. The primary innovation that we describe here is the OverFlap PCR principle. In our setting we combined it with whole plasmid amplification based mutagenesis, which was described by Laible & Boonrod already in 2009 [6] and employed for randomization by Galka et al. in 2017 [49]. Since our story covers all stages of methodological development from those described by Laible & Boonrod and Galka et al. to our final solution, we believe that our manuscript describes comparison to one of the “older methods” and clearly demonstrates improvements in the outcome. We also believe that, due to their large number and diversity, comparison of our approach to randomization methods, which are based on other types of procedures, would be impractical and selection of one method over another would always be controversial. 

 Comment: 3. What fraction of 18 amino acid sequence space is viable in E. coli AND S. cerevisiae? The authors focus exclusively on the biases of in vitro library construction, but it is quite possible that the biases (and limitations in transformation efficiency) imposed by host cells will preclude the discovery of functional peptides.

 Response: We are grateful for the interesting and relevant question and comments. In theory pool of 54 nt long random sequence oligonucleotides can yield up to 3.245×1032 sequence variants and at least 5.39×108 mol of oligo matter is required to ensure that all possible variants are represented, but due to fact that several codons code for the same aa residue such oligo pool would encode 6.31×1023 peptide variants. Thus, considering that our best library contained only 4 667 variants, the fraction of captured sequence variants is very small. However following that “cover all variants” logic it wouldn’t make sense to even consider randomizing anything beyond 8 codons (2.81×1014 variants), because one nmol of 24nt random sequence oligo matter would be required to cover the whole repertoire of possible variants. Even more prized high diversity libraries which are said to contain 1×109 variants are barely able to cover randomization of five codons. Therefore, we believe that instead of focusing on capture of all possible variants in single library, the decisions on employment of randomized libraries should be based on expected target occurrence probability estimations and library limitation assessment. For example, the pharmacophore of the same �-MSH is formed by four amino acid residue motif – His-Phe-Arg-Trp and the probability for occurrence of nucleotide sequence that encodes such motif in stretch of 12 nt (4 codon) long random oligo is 1:699 050.67. This means that, in contrast to full 12 nt (4 codon) long random oligo library with 1.68×107 variants, a 24 times smaller library with only 6.99×105 variants might be sufficient. Even more if the same calculation is carried out in the case of 54 nt (18 codon) long random oligo, the probability of such 4 aa residue motif occurrence somewhere in the sequence is even higher - 1:46 603.38, thus in theory the library of only 4.66×104 variants might suffice to ensure that the desired motif is present. As it can be seen from these calculations detection of motif that is similar to �-MSH pharmacophore can be achieved after implementation of envisioned methodological improvements (Gibson’s assembly) and when hunting for biologically active peptide motifs, randomization of longer sequence can actually improve the success rate. Regarding the exclusive focus on the biases, acquired data shows that employment of our main development -Asymmetric OverFlap PCR provides almost 20 fold improvement in terms of fraction of unique protein coding sequences per observed colonies (from 4.87% to 96.33%) and 2 fold improvement in fraction of randomized protein coding sequences (from 49.79% to 100%). Both of these parameters characterize the richness of our created libraries and their improvements are critical for acquisition of applicable random peptide library expression plasmid. 

 Comment: 4. Moreover, methods that link mRNA to their translation products in vitro (via ribosome display, encapsulation of mRNAs with working ribosomes, or a cross-linking puromycin analog) have produced large and diverse libraries. Those methods, however, have not been very widely adopted, in part because they are technically challenging but also because the results haven’t been as impressive as those produce by in silico design of de novo proteins. So bias in existing library construction methods might not be the bottleneck that the authors believe it to be.

 Response: We are thanking the reviewer for the valuable insight in repertoire of other methods that are able to link genotype to phenotype. However, as reviewer has already pointed out, all those methods are technically challenging, while PCR based approaches are often used due to their simplicity. We explained earlier that the main outcome of our study is the principle of Asymmetric OverFlap PCR, which provides ~ 40 fold improvement in the diversity of acquired library (20 fold increase in fraction of unique sequences × 2 fold increases in fraction of unbiased sequences). Although we agree with the reviewer and acknowledge this in the Discussion section that the bottle neck of the whole methodological pipeline in its current iteration is the circularization and transformation stage, we disagree that uncovered bias in existing library construction methods is somewhat irrelevant. In our experience majority of methodological pipelines in their initial iterations display multiple-successive bottle neck issues, which often exert devastating cumulative effect on the outcome, and, although bias problem might seem dwarfed by representation of 4667 out of 6.31×1023 possible peptide coding sequence variants, our data clearly shows that without application of asymmetric OverFlap PCR, acquired library would be essentially useless. In addition it is our strong belief that methodological pipelines are developed by whole scientific community and innovation in every element deserves to be published so that whole process can be continuously improved. Although we have our own future plans for employment of created libraries (as explained in our manuscript), we chose to present the OverFlap PCR approach in a separate article to increase its visibility. In our view presentation in a single article with those future results would attract only readers of specific field while those from other would miss this development.

 Comment: 5. Others have shown that the cloning efficiency of whole plasmid amplification products is inversely proportional to length. So the methods described here cannot be adapted to every application.

 Response: We are grateful for interesting remark and we agree with the reviewer that there are reports on negative linear correlation between molecular mass of the plasmid and cloning efficiency. However, since transformation efficiency is defined as the number of cfu per 1 µg of plasmid, we believe that the explanation to this is phenomena is more related to negative linear correlation between molecular weight of plasmid and number of its molecules in a set unit of mass, rather than decreased ability of larger molecule to enter the cell. In our view the same principle also applies to observed negative linear correlation between length of whole plasmid amplification product and cloning efficiency, because capacity of employed PCR setup defines the total yield of product in mass units and hence the number of acquired amplicons is negatively correlated with their size, thus, in the case of larger amplicons, lower number of molecules is used in transformation. Although frustrating, this shortcoming can be easily countered by increasing the volume of PCR mixture and subsequent concentration of amplification products during purification step. We also agree with the reviewer that the methods described in our manuscript cannot be adapted to every application, but, in our experience, this is the general state of all methods and only the reader can decide whether the current method is applicable to his/hers study. 

 Comment: 6. The numbers of colonies analyzed (<5000, Table 2) was miniscule compared to the number of possible sequences (4^54) or the number of NGS reads. Why produce such diverse libraries (or apply such high throughput analytic techniques) if transformation efficiency imposes such a tight bottleneck? The authors believe that the problem can be overcome via Gibson Assembly but it is probable that the transformation protocol (heat shock, growth of colonies on agar plates) is limiting.

 Response: We thank reviewer for this valuable comment. As we explained in response to comment No.4, the randomization of longer stretches of DNA sequence can increase the success rate when searching for biologically active aa sequence motifs, but, since further employment of acquired library is not the subject of this manuscript, we chose not to elaborate on it. However there is another aspect that favors selection of longer randomization sequences in presented type of study. As justly indicated by the reviewer, in theory randomization of 54 nucleotides long sequence may produce libraries with extremely high diversity. Although this theoretical level of diversity is unreachable in practice, employment of high diversity oligo pool improved our ability to detect randomization bias, by decreasing the probability that sequence similarity between two sequencing reads is coincidental. Our choice for application of high throughput analytic techniques was based solely on financial and workload aspects. The cost for sequencing of whole random peptide expression library employing available NGS technology was below €200 and the generation of sequencing library involved only PCR and following magnetic bead based purification steps, while the cost for Sanger sequencing of individual variants within all colonies would amount to ~€ 15000 and innumerable workhours. We also agree with the reviewer that chemical transformation protocol might be one of the bottle necks and many researchers have demonstrated that better results are achieved by electroporation. Therefore we have accordingly altered manuscript to incorporate this suggestion.

Comment: 7. The calculation of cumulative occurrence probability (COP) ought to be spelled out so that all readers (including this reviewer) can understand it.

Response: We appreciate the comment of the reviewer. Not being the native English speakers, we were are aware that there might be some grammar issues in our manuscript, therefore we double checked the spelling of “cumulative occurrence probability (COP)” everywhere within the manuscript, but unfortunately were unable to identify any spelling related issue. Thus we presumed that reviewer was either referring to incorrectness in choice of our terminology or to insufficiency of explanations on this subject. To comply with the first possibility we decided to replace selected term “cumulative occurrence probability (COP)” with the alternative: “Cumulative probability of occurrence (CPO)” and we hope that reviewer will find this change as acceptable. To comply with the second possibility, we added additional explanations that describe assumptions and calculations behind CPO and these changes will satisfy the reviewer.

Responses to Reviewer #2 Comments: 

 Comment: The manuscript "OverFlap PCR-a reliable approach for generation of plasmid DNA libraries containing random sequence without template bias" describes a straightforward method for generating randomized protein sequences without template bias, which is common and problematic with current PRC based methods. It's clear that the authors are experts in PCR techniques and have a strong understanding of the current issues with template bias in the pursuit of obtaining fully randomized sequences and the importance of this to identifying new therapeutic candidates. I feel this is an important paper that could have a significant impact in the quest to obtain fully randomized sequence with a relatively simple approach.

 Response: We are grateful for the kind words and we hope that reviewer will find the improved version of the manuscript even more appealing.

 Comment: 1. The manuscript is technically sounds with data supporting their conclusions. There are a few comments I would like to have addressed.

a. Figure 2 is a little confusing. The figures focus is asymmetric PCR, but the schematic shows both primers binding to the template. Perhaps, first show the overFlap primer binding to the template by itself, then below that, show the template again but this time with the addition of the sequence primer.

 Response: We are thanking reviewer for the valuable suggestion. Indeed presenting asymmetric PCR separately form following amplification improves perception of presented information. We have implemented suggested change in our figures.

 Comment: b. the labeling of the primers is confusing. Based on figure 1 (circular plasmid) and Figure 2, it appears to me that the "overFlap" primer is the forward primer, but in the texts and in table 1 its called the reverse primer. What appears to be the reverse primer in the figure is called the forward primer. Perhaps this is based on the primer location shown on the linearized plasmid in figure 1. However, based on all the other diagrams and the bulk of this work...this labeling is confusing.

 Response: We appreciate reviewer’s valuable insight. In our opinion designations “Forward” and “Reverse” are relevant only in the context of peptide coding sequences or whole genes, because both of these elements are located on one of the DNA chains, while when it comes to general PCR, whole plasmids or chromosomes these designations are a matter of involved researcher’s perspective and sometimes even a matter of chance. During creation of figures we tried to follow the pattern of normal English reading order (top to bottom – left to right), because readers usually first look at the elements that are located on the left side of the figure and consequently are paying more attention to those. However we see the reviewer’s point and agree that in current manuscript placing the OverFlap primer on the right side of the figure would be more in line with the whole story and less confusing to reader. This suggestion was implemented in all relevant figures.

 Comment: c. In Figure 4, its reported that plate C has the most colonies, however, visually, it doesn't appear to me to be the case. It appears to have the fewest colonies.

 Response: Yes, we agree with the reviewer that visually plate C appears to have lower number of colonies than others and we were surprised that in contrary to these observations employed “Open CFU” software reported larger number of colonies on this plate. However after checking all the analysis and detected colony parameters and even visual masks that marked detected colonies we concluded that false-negative and false-positive detection rate was very low and similar in all three plates. Our best explanation to this visual effect is that instinctively we are paying more attention to the center of the object and disregarding the periphery and putting greater emphasis on larger objects rather than smaller, but in the case of plate C the colonies are smaller than those on other plates and while their density is lower in the central part, it is a lot higher on the periphery. In essence this situation illustrates exactly the reason why we chose to employ software for counting of colonies rather than visual estimation.

 Comment: 2. The authors appear to have expertise on the mathematical calculations. However, I had a hard time following their math/logic, especially as it relates to the explanation of their approach to calculating cumulative occurrence probability (COP). For example, how did you arrive at the value for "probability of each variant occurrence"? The description related to COP needs to be more more clear.

 Response: We thank reviewer for kind words and indication of this shortcoming. We have elaborated on how calculations were carried out in the latest version of the manuscript. Also, in response to comments from another reviewer we decided to replace selected term “cumulative occurrence probability (COP)” with the alternative: “Cumulative probability of occurrence (CPO)”. We hope that these elaborations will be sufficient for the readers to understand performed calculations.

 Comment: 3. It appears the authors have made available all their data related to this work.

 Response: We thank reviewer for noting this and we agree that all relevant data has been provided.

 Comment: 4. This manuscripts need a lot of work with the writing. My biggest issue with the paper is that it is not clearly written. It has many typos, dropped words, and poorly constructed sentences. I recommend having this manuscript thoroughly edited by someone who speaks native English or at least has a stronger command of the English written language. There are too many typos and grammatical errors for me to address individually. However, I do want to highlight a couple issues.

 Response: We thank reviewer for this comment. Professional English editing has been carried out to improve its readability

 Comment: a. Page 12 under heading "Randomization employing modified...", second paragraph you wrote a couple times that "9 nucleotides where the first thymidine was replaced deoxy uridine". However, in Table 1 (and on page 8), you show 6 nucleotides were added prior to the deoxy uridine.

 Response: We are grateful that reviewer noticed this mistake. The correct number is 6 and according changes were made to manuscript. 

 Comment: b. Figure 8 and its detailed explanation is awkward and out of place in the discussion. It also seem a bit redundant to the earlier figures. I do think it is very useful information. I suggest you consider moving this to supplemental data.

 Response: We appreciate the valuable suggestion and we agree that moving this part to supplementary material section has improved the manuscript.

---

## [Decision Letter · Decision Letter 1]

26 May 2022

PONE-D-21-40826R1OverFlap PCR: A reliable approach for generating plasmid DNA libraries containing random sequences without a template biasPLOS ONE

Dear Dr. Davids Fridmanis, 

Thank you for submitting your manuscript to PLOS ONE. After careful consideration, we feel that it has merit but does not fully meet PLOS ONE’s publication criteria as it currently stands. Therefore, we invite you to submit a revised version of the manuscript that addresses the points raised during the review process.

ACADEMIC EDITOR: There are few comments from the reviewers need response from the authors

We look forward to receiving your revised manuscript.

Kind regards,

Shawky M Aboelhadid, PhD

Academic Editor

PLOS ONE

Journal Requirements:

Reviewers' comments:

Reviewer's Responses to Questions

**Comments to the Author**

1. If the authors have adequately addressed your comments raised in a previous round of review and you feel that this manuscript is now acceptable for publication, you may indicate that here to bypass the “Comments to the Author” section, enter your conflict of interest statement in the “Confidential to Editor” section, and submit your "Accept" recommendation.

Reviewer #1: (No Response)

Reviewer #2: All comments have been addressed

2. Is the manuscript technically sound, and do the data support the conclusions?

Reviewer #1: Yes

Reviewer #2: Yes

3. Has the statistical analysis been performed appropriately and rigorously? 

Reviewer #1: I Don't Know

Reviewer #2: Yes

4. Have the authors made all data underlying the findings in their manuscript fully available?

Reviewer #1: Yes

Reviewer #2: Yes

5. Is the manuscript presented in an intelligible fashion and written in standard English?

Reviewer #1: No

Reviewer #2: Yes

6. Review Comments to the Author

Reviewer #1: Fridmanis et al. revised their manuscript. The new version is easier to understand, but would benefit from further revision to clarify the following points of confusion.

1. The authors have created a diverse library of secreted peptides. Won’t secretion sever the link between genotype and phenotype? How would they identify clones with biological function?

2. How is cumulative occurrence probability (COP), now calledk CPO, computed?

Reviewer #2: Authors have addressed my concerns and made other changes that feel improved the manuscript. The changes the authors made to the figures improved clarity of the technique and the re-write in explaining the rational behind CPO was helpful. I feel the manuscript is technically and grammatically sound and of value.

7. PLOS authors have the option to publish the peer review history of their article (what does this mean?). If published, this will include your full peer review and any attached files.

Reviewer #1: No

Reviewer #2: No

---

## [Author Response · Author response to Decision Letter 1]

14 Jul 2022

Responses to Reviewer #1 Comments:

Comment: 1. The authors have created a diverse library of secreted peptides. Won’t secretion sever the link between genotype and phenotype? How would they identify clones with biological function?

Response: We are grateful for the Reviewer’s insight and concern. We agree that maintaining the link between genotype and phenotype is a major issue in some applications of random peptide expression libraries, especially in those that involve secretion. However application of created random peptide library expression plasmids is not the subject of this article. We provided marginal insight into our future plans only to explain why the assessment of randomization process efficiency was carried out on the encoded amino acid sequence level and not on nucleic acid sequence level, but our research activities are being continued and if the reviewer would be so kind to reveal him/her self by providing the e-mail address, we will be happy to send the link to the preprint of our next article as soon as it will be finished and made available to public.

Comment: 2. How is cumulative occurrence probability (COP), now calledk CPO, computed?

Response: We thank Reviewer for the comment and we understand the confusion that this change in terminology might have caused. However we have already explained the reasons behind these changes in our response to one of reviewer’s previous comments (Comment No. 7), therefore we decided to abstain from provision of essentially identical information, but for the convenience of review process we are including both previous comment and its response as an inline element of our current response: 

Reviewer’s previous Comment: “7. The calculation of cumulative occurrence probability (COP) ought to be spelled out so that all readers (including this reviewer) can understand it.” 

Our previous Response: “We appreciate the comment of the reviewer. Not being the native English speakers, we were are aware that there might be some grammar issues in our manuscript, therefore we double checked the spelling of “cumulative occurrence probability (COP)” everywhere within the manuscript, but unfortunately were unable to identify any spelling related issue. Thus we presumed that reviewer was either referring to incorrectness in choice of our terminology or to insufficiency of explanations on this subject. To comply with the first possibility we decided to replace selected term “cumulative occurrence probability (COP)” with the alternative: “Cumulative probability of occurrence (CPO)” and we hope that reviewer will find this change as acceptable. To comply with the second possibility, we added additional explanations that describe assumptions and calculations behind CPO and these changes will satisfy the reviewer.”

Responses to Reviewer #2 Comments: 

Comment: Authors have addressed my concerns and made other changes that feel improved the manuscript. The changes the authors made to the figures improved clarity of the technique and the re-write in explaining the rational behind CPO was helpful. I feel the manuscript is technically and grammatically sound and of value.

Response: Once again we are grateful for the reviewer’s kind words and we are happy that improvements of the manuscript were found suitable for publication .

---

## [Decision Letter · Decision Letter 2]

18 Jul 2022

OverFlap PCR: A reliable approach for generating plasmid DNA libraries containing random sequences without a template bias

PONE-D-21-40826R2

Dear Dr. Davids Fridmanis, 

We’re pleased to inform you that your manuscript has been judged scientifically suitable for publication and will be formally accepted for publication once it meets all outstanding technical requirements.

Kind regards,

Shawky M Aboelhadid, PhD

Academic Editor

PLOS ONE

Additional Editor Comments (optional):

Reviewers' comments:

Reviewer's Responses to Questions

**Comments to the Author**

1. If the authors have adequately addressed your comments raised in a previous round of review and you feel that this manuscript is now acceptable for publication, you may indicate that here to bypass the “Comments to the Author” section, enter your conflict of interest statement in the “Confidential to Editor” section, and submit your "Accept" recommendation.

Reviewer #1: (No Response)

2. Is the manuscript technically sound, and do the data support the conclusions?

Reviewer #1: Yes

3. Has the statistical analysis been performed appropriately and rigorously? 

Reviewer #1: Yes

4. Have the authors made all data underlying the findings in their manuscript fully available?

Reviewer #1: Yes

5. Is the manuscript presented in an intelligible fashion and written in standard English?

Reviewer #1: Yes

6. Review Comments to the Author

Reviewer #1: The authors didn't address either of my concerns to my satisfaction. Some readers won't understand the utility of a library of random sequence peptides unlinked to the encoding sequences. And most won't understand how the authors computed CPO. It is nevertheless possible that the published article will motivate some readers to think harder about the biases within their own libraries. To the extent that happens the article will be a constructive contribution to the field.

7. PLOS authors have the option to publish the peer review history of their article (what does this mean?). If published, this will include your full peer review and any attached files.

Reviewer #1: No

---

## [Editor Report · Acceptance letter]

28 Jul 2022

PONE-D-21-40826R2 

OverFlap PCR: A reliable approach for generating plasmid DNA libraries containing random sequences without a template bias 

Dear Dr. Fridmanis:

I'm pleased to inform you that your manuscript has been deemed suitable for publication in PLOS ONE. Congratulations! Your manuscript is now with our production department. 

Kind regards, 

on behalf of

Professor Shawky M Aboelhadid 

Academic Editor

PLOS ONE